

# Evolution of inbreeding: a gaze into five Italian beef cattle breeds history

Giacomo Rovelli[1,2], Maria Gracia Luigi-Sierra[2], Dailu Guan[2,3], Fiorella Sbarra[4], Andrea Quaglia[4], Francesca Maria Sarti[1], Marcel Amills[2,5] and Emiliano Lasagna[1]

[1] Department of Agricultural, Food and Environmental Sciences (DSA3), University of Perugia, Perugia, Italy
[2] Centre for Research in Agricultural Genomics (CRAG), CSIC-IRTA-UAB-UB, Universitat Autónoma de Barcelona, Bellaterra, Barcelona, Spain
[3] Department of Animal Science, University of California, Davis, Davis, CA, United States of America
[4] National Association of Italian Beef-Cattle Breeders (ANABIC), San Martino in Colle, Perugia, Italy
[5] Departament de Ciència Animal i dels Aliments, Universitat Autónoma de Barcelona, Bellatera, Barcelona, Spain

## ABSTRACT

In the last decades, intensive selection programs have led to sustained increases of inbreeding in dairy cattle, a feature that might have adverse consequences on the viability and phenotypic performance of their offspring. This study aimed to determine the evolution of inbreeding of five Italian beef cattle breeds (Marchigiana, Chianina, Romagnola, Maremmana, and Podolica) during a period of almost 20 years (2002–2019). The estimates of $Ho$, $He$, $F_{hat2}$, and $F_{ped}$ averaged across years (2002–2019) in the studied breeds fluctuated between 0.340–0.401, 0.348–0.392, –0.121–0.072, and 0.000–0.068, respectively. Moreover, annual rates of increase of the estimated inbreeding coefficients have been very low ($F_{hat2}$ = 0.01–0.02%; $F_{ped}$ = 0.003–0.004%). The use of a high number of bulls combined with strategies implemented by the Association of Italian Beef Cattle Breeders ANABIC to minimize inbreeding might explain these results. Despite the fact that diversity and inbreeding have remained quite stable during the last two decades, we have detected a sustained decrease of the population effective size of these five breeds. Such results should be interpreted with caution due to the inherent difficulty of estimating $N_e$ from SNPs data in a reliable manner.

# INTRODUCTION

Inbreeding is the main consequence of mating individuals that are related, through common ancestry, to a degree that exceeds that of two individuals from the same population extracted at random (*Kardos et al., 2016*). Minimizing inbreeding is an aspect of paramount importance in cattle breeding to avoid the phenotypic expression of detrimental alleles in the offspring as well as to ensure the maintenance of genetic diversity (*Howard et al., 2017*). Inbred animals display chromosome segments that are identical-by-descent (IBD) and generate long runs of homozygosity (*Kardos et al., 2016*). Although inbreeding coefficients can be calculated from pedigrees comprising several generations, in recent years the advent of high throughput genotyping techniques has make it possible to estimate

Corresponding authors
Marcel Amills, marcel.amills@uab.cat
Emiliano Lasagna, emiliano.lasagna@unipg.it

molecular inbreeding coefficients based on the characterization of the genome-wide patterns of homozygosity. One key advantage of this latter approach is that it captures ancient inbreeding accumulated in the base population and it is less affected by parentage errors (*Howard et al., 2017*). However, several studies (*Bjelland et al., 2013*; *Pryce et al., 2014*) have revealed the distortion caused by ancient contributions to inbreeding dilutes up to a degree in which pedigree-based analyses and genomic analyses may not differ that much.

Inbreeding is often reported in domestic animal populations as a measurement of one or several coefficients of inbreeding at a particular time point. However, assessing the magnitude of inbreeding on a continuous temporal scale is much more informative because it captures its tendency and predicted behaviour. Between 1960 and 2000, inbreeding coefficients of US dairy breeds, such as Ayrshire, Brown Swiss, Guernsey, Holstein, and Jersey, went from 0% (base population) to 4.5–6% in just four decades (*Weigel, 2001*). According to *Weigel (2001)*, this increase in inbreeding was not associated with effective population size and, more likely, it was the result of the intensity of genetic selection as well as of the extensive use of a reduced number of elite sires. Similarly, *Mc Parland et al. (2007)* investigated the evolution of inbreeding in the Charolais, Limousine, Hereford, Angus, and Simmental beef cattle breeds as well as in the Holstein-Friesian dairy cows raised in Ireland. They found that, between 1960 and 2004, overall inbreeding increased from 0.10–0.25% to 0.5–2%, and over the last decade (1994–2004) the annual rate of increase in inbreeding was 0.06–0.13%. These and other studies suggest that inbreeding is accumulating rapidly in cattle breeds due to efficient genetic selection programs and reproductive management (*Weigel, 2001*).

Genetic selection in Italian beef cattle is implemented by the National Association of Italian Beef Cattle Breeders (ANABIC) and aimed to improve meat production, precocity, growth ability, and muscle development (*Sbarra, 2011*). Three of the five main Italian beef cattle breeds, Marchigiana (MAR), Chianina (CHI), and Romagnola (ROM) are highly specialised in beef production, and the other two, Maremmana (MRM) and Podolica (POD), are considered as rustic (*Sarti et al., 2019*). The current selection program, based on the traditional quantitative approach, has achieved a remarkable improvement of growth, daily weight and muscularity gain (*Sbarra, 2011*). Moreover, cattle are somatically well-developed with a correct morphology and light skeletal system (*Rovelli et al., 2020a*). However, the intensity of the selection in the five breeds is lower in the rustic MRM and POD than in the three specialised ones; moreover, the two rustic breeds register a low amount of young bulls/year in performance test (*Fioretti et al., 2020*). In the current work, we aimed to characterize the historical evolution of inbreeding in five Italian beef cattle breeds (CHI, MAR, ROM, MRM, and POD) in the period comprised between 2002 and 2019 by using both molecular and genealogical estimates of inbreeding coefficients. Our goal was to test whether intensive selection performed in the last twenty years has resulted in a significant increase of inbreeding levels in these five populations.

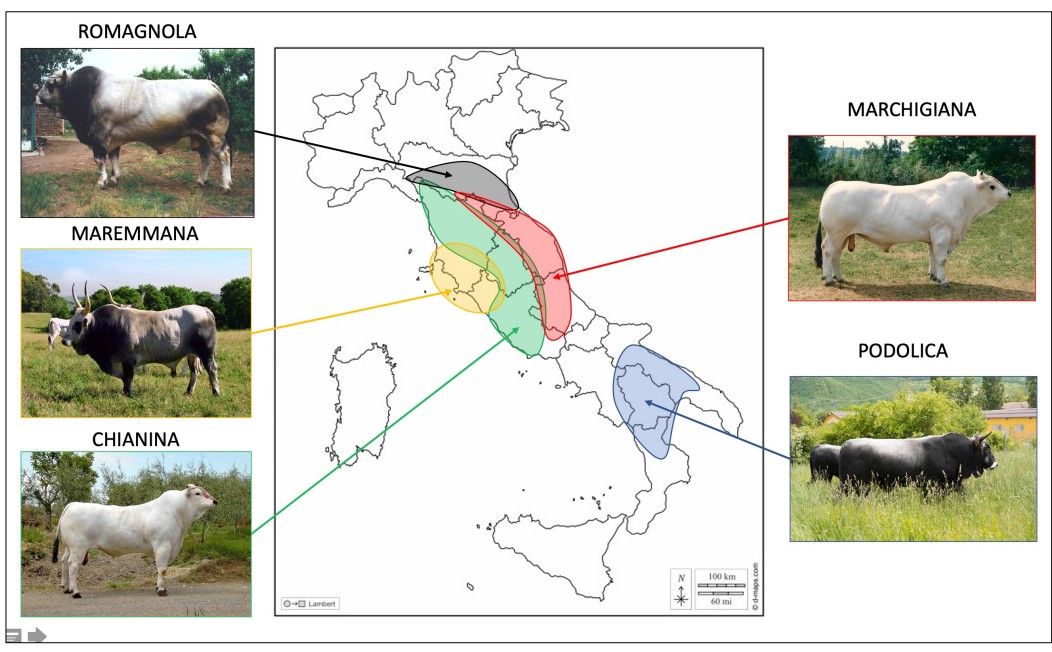

**Figure 1** **Geographical spread of the studied breeds in the different Italian regions.** Photo credit: AN-ABIC. Map of Italy adapted from https://www.d-maps.com/carte.php?num_car=14531&lang=it.

## MATERIALS & METHODS

### Cattle sampling and genotyping

This work comprised 3,581 young bulls belonging to five of the main Italian beef cattle breeds: CHI (909), MAR (879), ROM (904), MRM (334), and POD (555). The number of young bulls and their year of birth are reported in Table S1. Samples were collected by staff from the ANABIC at the genetic station of San Martino in Colle (Perugia, Italy) during 1985–2019. Geographical distribution of these breeds in Italy and the pictures of representative individuals from each breed can be found in Fig. 1.

During the performance test, blood samples were taken from the jugular veins of each bull. These samples were collected in EDTA $K_3$ coated vacuum tubes and stored at −20 °C (*Rovelli et al., 2020b*). Genomic DNA was purified with the GenElute Blood Genomic DNA kit (Sigma Aldrich, St. Louis, MO, USA). The purification method was previously described in *Sarti et al. (2019)*. Genomic DNA samples from the 3,581 bulls were genotyped with the GeneSeek Genomic Profiler Bovine LDv4 33K chip (Illumina Inc., San Diego, CA, USA), which contains 30,111 SNPs, and processed at the Agrotis laboratory (LGS, Cremona, Italy). Standard multi-sample protocols and reagents were used according to the instructions of the manufacturer (*Khatkar et al., 2012*).

The positions of single nucleotide polymorphisms (SNPs) were mapped using the ARS-UCD_1.2 bovine genome assembly (*Zorc, Ogorevc & Dovc, 2019*). The PLINK software v1.9 (*Chang et al., 2015*) was used to update the names and positions of SNP markers. The SNPs that did not match the following criteria were removed before performing

population diversity analyses: (i) SNPs with call rates less than 90%, (ii) SNPs with minor allele frequencies less than 5%, (iii) SNPs with more than 1% missing genotypes, and (iv) SNPs displaying highly significant deviations ($P$-value $< 10^{-3}$) from the Hardy-Weinberg equilibrium (*Amaral, Pavão & Gama, 2020*). Another pruning step was performed to remove SNPs with high linkage disequilibrium (LD) using the command –indep 50 5 2 (*Manunza et al., 2016*) of the PLINK v1.9 software (*Chang et al., 2015*) as recommended in a previous publication (*Howrigan, Simonson & Keller, 2011*). This second step is necessary because stretches of SNPs with low MAF and genomic regions with many SNPs and strong LD often yield erroneous estimates of the effective population size (*Manunza et al., 2016*). The results of these pruning steps are shown in Table S2.

## Data analysis
### Calculation of molecular inbreeding coefficient
In the five studied breeds, inbreeding coefficient $F_{hat2}$ was calculated for each bull with the PLINK v1.9 software (*Chang et al., 2015*). The –ibc command of PLINK v1.9 (*Chang et al., 2015*) was used to compute $F_{hat2}$. The formula used to calculate $F_{hat2}$ is as follows:

$$F_{hat2} = \frac{O_{hom} - E_{hom}}{1 - E_{hom}}$$

where $O_{hom}$ is the observed number of homozygotes and $E_{hom}$ is the expected number of homozygotes.

The mean of the inbreeding coefficient was calculated per year (18 levels) for each one of the five studied breeds. Animals born between 1985 and 2002 were merged into a single group because the number of genotyped individuals born before 2002 is very scarce. The PROC REG v14.1 tool (SAS Inst. Inc., Cary, NC) was used to estimate the annual rate of increase in $F_{hat2}$ by fitting a linear regression and considering the 2002–2019 period (*Sall, 1981*). The R software v4.0.3 (*R Core Team, 2018*) was used to perform a box plot to represent graphically the inbreeding coefficient variation per year for each of the studied breeds.

### Calculation of a pedigree –based inbreeding coefficient
The Endog software v4.8 (*Gutiérrez & Goyache, 2005*; *Gutiérrez, Goyache & Cervantes, 2009*) was used to calculate the pedigree inbreeding coefficient ($F_{ped}$), which is defined as the probability that an individual has two IBD alleles (*Ferenčaković et al., 2017*). In addition to the default variables proposed by the software, we also considered for $F_{ped}$ estimation the average relatedness (AR) coefficient. This parameter is defined as the probability of an allele, chosen randomly from the entire population, to belong to a given animal, so AR can be understood as the representation of the animal in the entire pedigree regardless of the knowledge of such pedigree (*Gutiérrez, Goyache & Cervantes, 2009*).

The depth and completeness of the pedigree are key when estimating inbreeding coefficients, because an incomplete pedigree will lead to an underestimation of the mean inbreeding. We calculated a pedigree completeness index (PCI) for each animal included in the pedigree using the method developed by *MacCluer et al. (1983)* and implemented in *Sargolzaei (2014)*. The depth of the pedigree varied across breeds, since we have considered
only the generations with all known ancestors. For the rustic breeds (MRM and POD), the inbreeding coefficient ($F_{ped}$) was calculated considering four ancestral generations. In contrast, the MAR and CHI breeds were represented by individuals from three ancestral generations while in ROM only genealogical data from two generations were available. As previously said, the mean $F_{ped}$ per year was computed for each of the five studied breeds, merging in one single group the animals born between 1985 and 2002. The annual rate of increase in $F_{ped}$ coefficient was estimated by fitting a linear regression using PROC REG v14.1 (SAS Inst. Inc., Cary, NC) through the time period from 2002 to 2019 (*Sall, 1981*). This linear regression was plotted with the R software v4.0.3 (*R Core Team, 2018*).

### Estimation of genetic diversity and historic effective population size trends

The Arlequin software v3.5.2.2 (*Excoffier & Lischer, 2010*) was used to estimate within-population diversity, by calculating observed (*Ho*) and expected (*He*) heterozygosities subsequently corrected over the number of usable SNPs.

Historical trends in effective population size ($N_e$) were estimated with the SNeP software (*Barbato et al., 2015*) using default settings and a correction to adjust linkage disequilibrium (LD) $r^2$ values for small sample sizes. The same index was also calculated through the individual increase in inbreeding, using the software Endog v4.8 (*Gutiérrez & Goyache, 2005*; *Gutiérrez, Goyache & Cervantes, 2009*).

The formula used to estimate $N_e$ from LD (*Corbin et al., 2012*), with SNeP software, was:

$$N_{T(t)} = \frac{1}{(4f(c_t))}\left(\frac{1}{E\left[r^2_{adj}|C_t\right]} - \alpha\right)$$

Where $N_{T(t)}$ is the effective population size estimated $t$ generations ago in the past, $c_t$ is the recombination rate $t$ generations ago in the past, $r^2_{adj}$ is the linkage disequilibrium estimation adjusted for sampling bias, and $\alpha$ is a constant.

## RESULTS

The average PCI from 2002 to 2019 ranged from 99.29 to 99.91% (MAR), from 99.17 to 99.89% (CHI), from 98.80 to 99.92% (ROM), from 99.01 to 99.87% (MRM), and from 99.02 to 99.86% (POD). The removal of genotyped animals with PCI less than 90% resulted in the exclusion of less than 1.5% of the sample, as most of the genotyped animals had PCI greater than 90%. The estimates of *Ho*, *He*, $F_{hat2}$, and $F_{ped}$ averaged across years (2002-2019) in the studied breeds fluctuated between 0.340–0.401, 0.348–0.392, −0.121–0.072, and 0.000–0.068, respectively (Tables 1 and 2). The $F_{ped}$ coefficients were higher in the rustic POD and MRM breeds, probably because the depth of the pedigree (in our dataset) is higher than in the MAR, CHI, and ROM breeds. Moreover, POD was the breed that displayed the highest $F_{hat2}$ coefficient, followed by ROM. The analysis of the evolution of $F_{hat2}$, $F_{ped}$, and *Ho* in the five breeds (Figs. 2 and 3) evidenced that the observed heterozygosity, in the five studied populations, remained constant throughout the years, with slightly higher values in the rustic breeds. With regard to $F_{hat2}$ and $F_{ped}$, we observed
**Table 1** Mean, standard deviation, and range (minimum and maximum values) for expected and observed heterozygosity.

| Breed | $Ho$[a] | | $He$[b] | |
|---|---|---|---|---|
| | $\bar{x} \pm sd$[c] | range[d] | $\bar{x} \pm sd$[c] | range[d] |
| Marchigiana | 0.350 ± 0.022 | 0.349–0.351 | 0.350 ± 0.019 | 0.349–0.352 |
| Chianina | 0.361 ± 0.024 | 0.359–0.362 | 0.356 ± 0.020 | 0.354–0.359 |
| Romagnola | 0.343 ± 0.021 | 0.340–0.345 | 0.356 ± 0.020 | 0.354–0.359 |
| Maremmana | 0.383 ± 0.025 | 0.380–0.384 | 0.391 ± 0.021 | 0.388–0.392 |
| Podolica | 0.399 ± 0.025 | 0.397–0.401 | 0.391 ± 0.021 | 0.388–0.392 |

**Notes.**
[a] $Ho$, observed heterozygosity
[b] $He$, expected heterozygosity
[c] $\bar{x} \pm sd$, mean and standard deviation of $Ho$ and $He$
[d] range, minimum and maximum values of $Ho$ and $He$.

**Table 2** Mean, standard deviation, and range (minimum and maximum values) for different inbreeding coefficients.

| Breed | $F_{hat2}$[a] | | $F_{ped}$[b] | |
|---|---|---|---|---|
| | $\bar{x} \pm sd$[c] | range[d] | $\bar{x} \pm sd$[c] | range[d] |
| Marchigiana | 0.012 ± 0.005 | −0.051–0.071 | 0.018 ± 0.002 | 0.002–0.048 |
| Chianina | 0.018 ± 0.005 | −0.051–0.023 | 0.024 ± 0.003 | 0.002–0.053 |
| Romagnola | 0.023 ± 0.010 | −0.121–0.072 | 0.007 ± 0.001 | 0.000–0.016 |
| Maremmana | 0.014 ± 0.003 | −0.061–0.044 | 0.062 ± 0.006 | 0.039–0.067 |
| Podolica | 0.025 ± 0.006 | −0.110–0.047 | 0.061 ± 0.006 | 0.032–0.068 |

**Notes.**
[a] $F_{hat2}$: molecular inbreeding coefficient.
[b] $F_{ped}$: pedigree-based inbreeding.
[c] $\bar{x} \pm sd$, mean and standard deviation of $F_{hat2}$ and $F_{ped}$.
[d] range: minimum and maximum values of $F_{hat2}$ and $F_{ped}$.

some fluctuations across years that were particularly accentuated for $F_{hat2}$ which showed a stable or increasing trend depending on the breed under consideration. In any case, these yearly oscillations in the magnitude of inbreeding were not very important. The MRM and POD breeds lacked data in one and three years, respectively. These missing values are due to the fact that in Southern Italy, between 2004–2006, there was a "Bluetongue" epidemia which caused a temporary cessation of the activities of the POD selection center. For the same reason, MRM selection was temporarily suspended in 2013. The annual rates of increase in inbreeding ($F_{hat2}$ and $F_{ped}$) are displayed in Figs. 4 and 5. It can be seen that in general $F_{hat2}$ increases slightly but steadily in all five breeds, with averaged overall increasing rates of 0.17–0.34% between 2002–2019. In contrast, $F_{ped}$ remained quite stable across time with averaged overall increasing rates of −0.04–0.08%. The effective population size ($N_e$) estimated t generations ago (from 13 to 235) is shown in Fig. 6. It is apparent that $N_e$ decreases markedly across generations. Thirteen generations ago, $N_e$ was lower than 300 for most breeds with the only exception of POD cattle ($N_e = 498$). In contrast, 235 generations ago $N_e$ oscillated between 1887–3257, which is 7.48 times larger than current values.

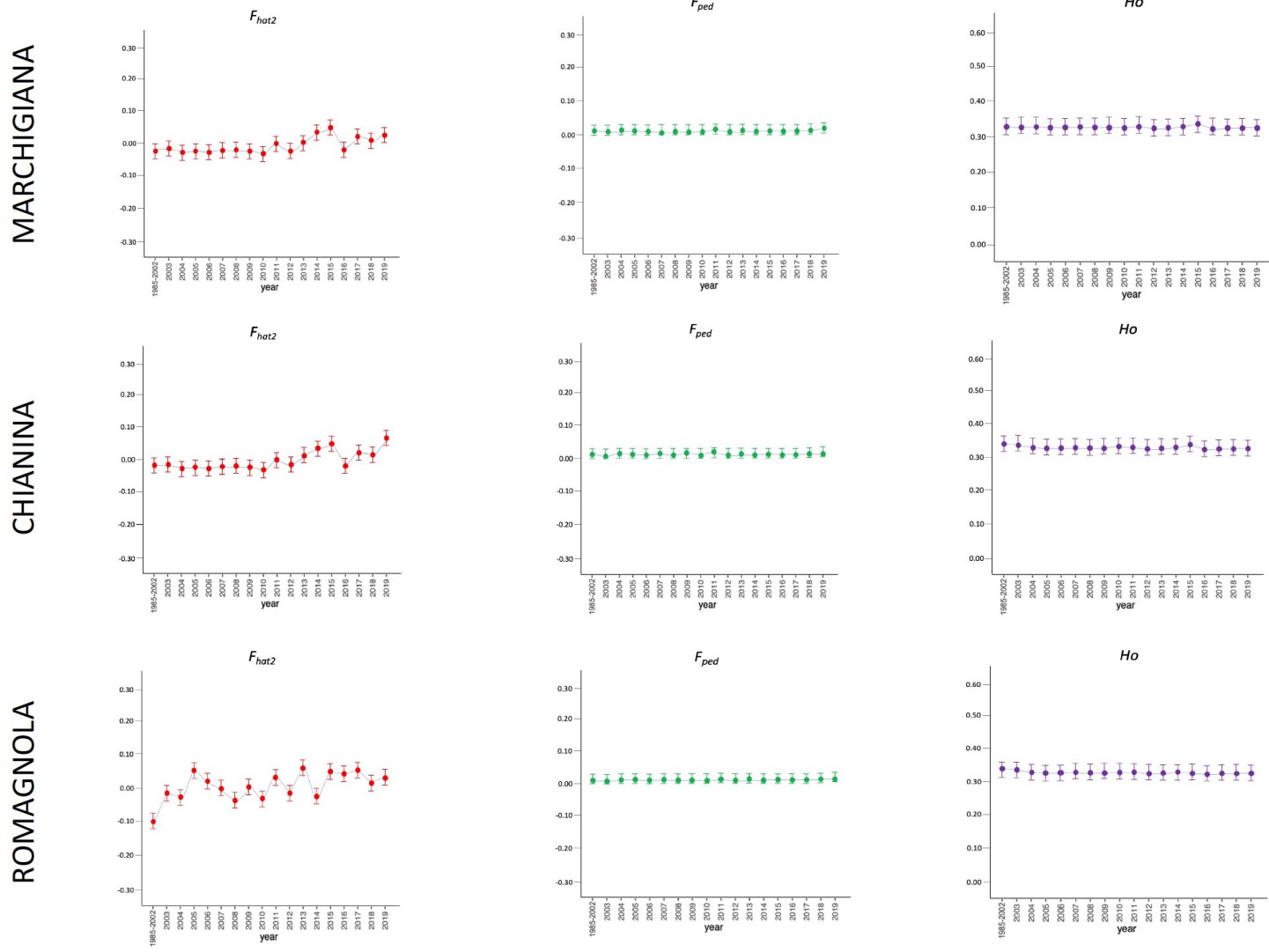

**Figure 2** Molecular ($F_{hat2}$) and pedigree ($F_{ped}$) inbreeding coefficients and observed heterozygosity of the specialised breeds (Marchigiana, Chianina, and Romagnola) measured in a period of almost 20 years (2002–2019). $Ho$: observed heterozygosity; $F_{hat2}$: molecular inbreeding coefficient; $F_{ped}$: pedigree-based inbreeding. The dots represent the mean value and the whiskers are the standard deviation. The base generation, represented as 1985–2002, is composed by animals born between 1985 and 2002 (they have been merged in a single group because data from few bulls were available before 2002).

## DISCUSSION

In this study, we have evaluated the variation of inbreeding and diversity parameters in five Italian beef cattle breeds across a window of approximately 20 years. We did not measure $F_{roh}$ because the number of SNPs was too small (21,000–23,000 valid SNPs) to map runs of homozygosity in a reliable manner. Importantly, in our study the base generation of each breed was composed of animals born before 2002. We did so because there was a very small number of bulls representing each one of the years comprised between 1985 and 2002, so they were merged together in a single group. We have observed that the

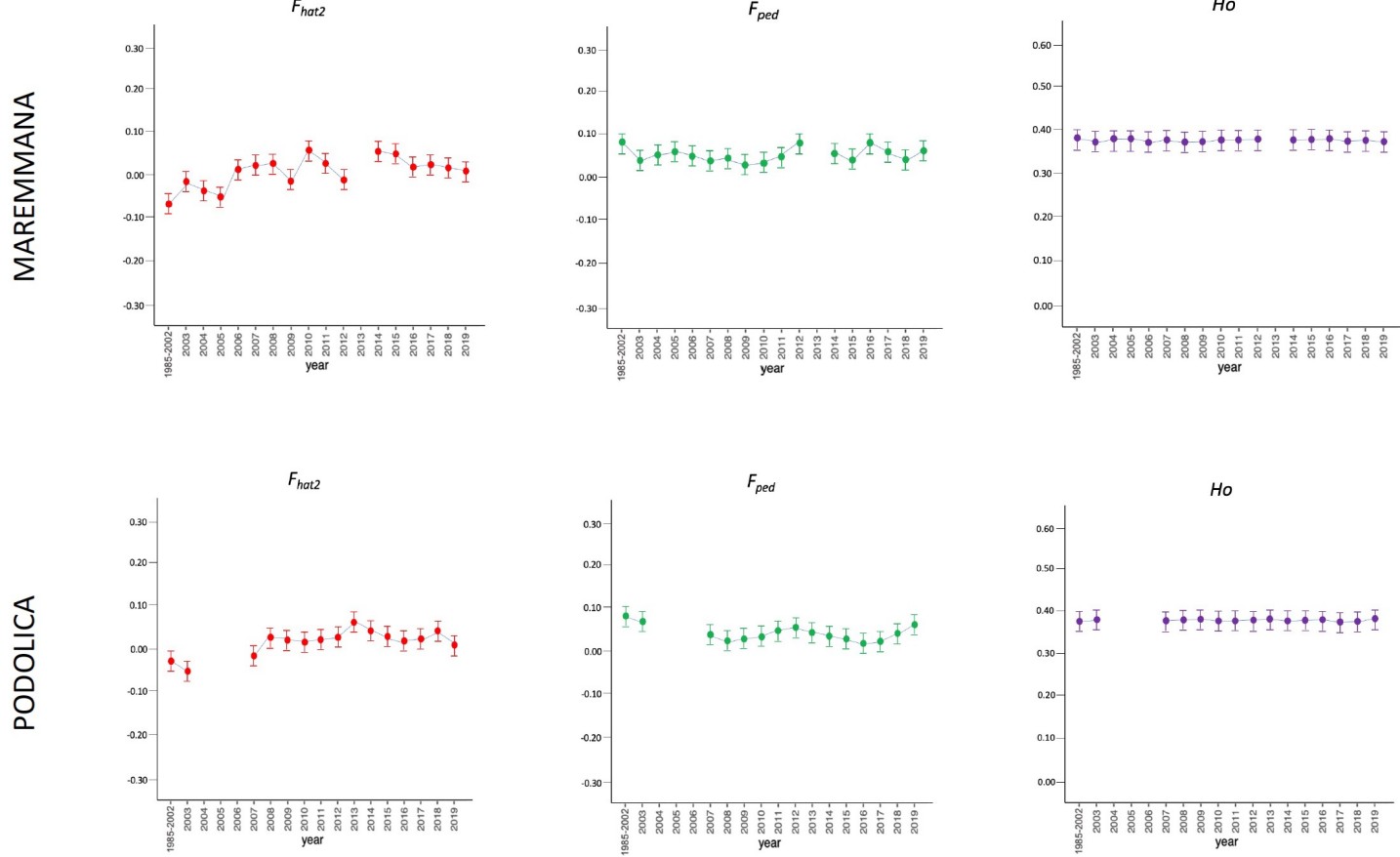

**Figure 3** Molecular ($F_{hat2}$) and pedigree ($F_{ped}$) inbreeding coefficients and observed heterozygosity of the rustic breeds (Maremmana and Podolica) measured in a period of almost 20 years (2002–2019). *Ho*: observed heterozygosity; $F_{hat2}$: molecular inbreeding coefficient; $F_{ped}$: pedigree-based inbreeding. The dots represent the mean value and the whiskers are the standard deviation. The base generation, represented as 1985–2002, is composed by animals born between 1985 and 2002 (they have been merged in a single group because data from few bulls were available before 2002).

averaged (across years) coefficients of inbreeding $F_{hat2}$ and $F_{ped}$ in the five Italian breeds are generally lower than 0.07 (Table 2, Figs. 2–5), while homozygosity was approximately 0.64–0.70 (data not shown). The $F_{hat2}$ coefficient is closely related to $F_{is}$ and can be interpreted (also in the case of $F_{ped}$ computed $F_{is}$) in breeding policy terms: in fact, positive and high values mean that mating between close relative are not –or cannot be –avoided. In smaller populations, like those at issue, breeders are implementing a more rigorous breeding policy to limit inbreeding within the herd. The MRM and POD breeds displayed the largest $F_{ped}$ coefficients but this was expected because the depth of the pedigree, in our study, is larger than for CHI, MAR, or ROM. Another reason explaining the larger $F_{ped}$ (and lower AR) observed in the rustic breeds (MRM and POD) relies on the fact that in these breeds artificial insemination (AI) is little spread. Bulls, especially in the past, were the offspring of animals born in the same farm. Furthermore, farms always remained quite isolated and poorly genetically connected to each other, at least until the foundation of test

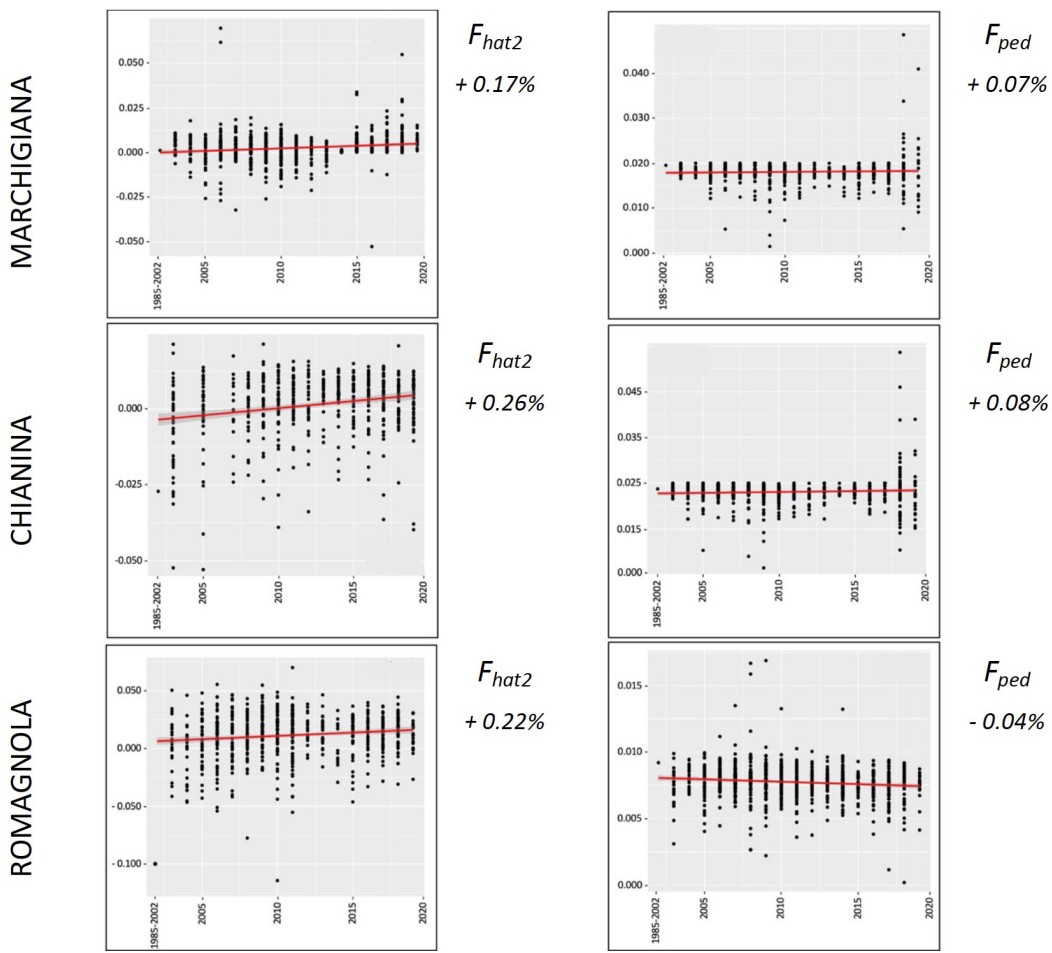

**Figure 4 Overall percentual increase of $F_{hat2}$ and $F_{ped}$ for the specialised breeds (Marchigiana, Chianina, and Romagnola).** $F_{hat2}$: molecular inbreeding coefficient; $F_{ped}$: pedigree-based inbreeding. The red line represents the regression of the coefficients over time expressed in years. The numbers next to the figure correspond to the overall increase of inbreeding across all analyzed years.

stations (*Moioli, Napolitano & Catillo, 2004*). On the other hand, in the specialized breeds (MAR, CHI, and ROM), the implementation of AI involved the use of unrelated lines thus avoiding inbreeding to a great extent (higher AR). We have observed slight discrepancies between $F_{hat2}$ and $F_{ped}$ values displayed in Table 2, but this outcome is probably explained by the fact that these two coefficients have different properties (*Alemu et al., 2021*). Indeed, $F_{ped}$ indicates the probability that two homologous alleles in an individual are identical by descent, as defined by *Malécot (1948)*, and it ranges from 0 to 1. In contrast, $F_{hat2}$ is very similar to the method-of-moments $F$ coefficient measured with the –het command of PLINK v1.9 (*Purcell et al., 2007*) which estimates the reduction of heterozygosity (or the excess of homozygosity) associated with inbreeding (*Alemu et al., 2021*). In consequence, $F_{hat2}$ can take negative values (*Alemu et al., 2021*). Despite these conceptual differences, comparison of $F_{ped}$ and $F_{hom}$ (similar to $F_{hat2}$) coefficients calculated in a pedigree of 245 Holstein cattle with whole-genome sequence data showed a good agreement between both

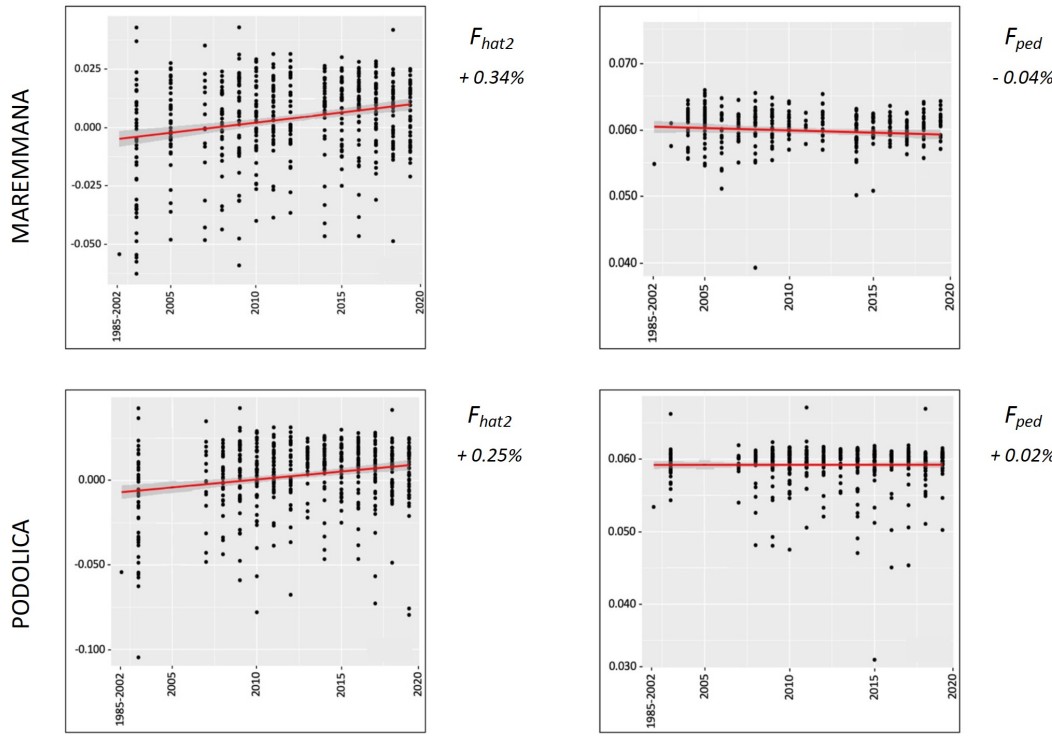

**Figure 5** **Overall percentual increase of $F_{hat2}$ and $F_{ped}$ for the rustic breeds (Maremmana and Podolica).** $F_{hat2}$: molecular inbreeding coefficient; $F_{ped}$: pedigree-based inbreeding. The red line represents the regression of the coefficients over time expressed in years. The numbers next to the figure correspond to the overall increase of inbreeding across all analyzed years.

parameters (*Alemu et al., 2021*). In our study, both coefficients indicate that none of the Italian breeds under study is significantly inbred.

We have also observed a lack of relationship between population size and the magnitude of inbreeding coefficients (Table 2, Figs. 2–5). For instance, MRM ($F_{hat2} = 0.014$, $F_{ped} = 0.061$) and ROM ($F_{hat2} = 0.023$, $F_{ped} = 0.023$) have population sizes of approximately 11,000–12,000 individuals, while MAR has a census almost five times larger but fairly comparable levels of inbreeding ($F_{hat2} = 0.012$, $F_{ped} = 0.018$). The amount of inbreeding is mostly determined by the demographic history of populations rather than by their current size. In local breeds undergoing strong demographic reductions, genetic drift can be quite intense thus increasing homozygosity and the occurrence of matings between related individuals. For instance, Chillingham cattle (a breed that lives in Northumberland earldom, England) are currently represented by a herd of 50 males and 50 females, which has remained reproductively closed in the last 300–350 years (*Visscher et al., 2001*). This herd was formed by five bulls and eight cows in 1947, and the average number of males and females per generation has been three and 15, respectively (*Visscher et al., 2001*). Calculation of $F_{is}$ inbreeding coefficient in Chillingham cattle yielded a value of 0.92, which is extremely high (*Williams et al., 2016*). In strong contrast, the population sizes of the five Italian breeds analyzed in the current work have remained relatively stable in the last

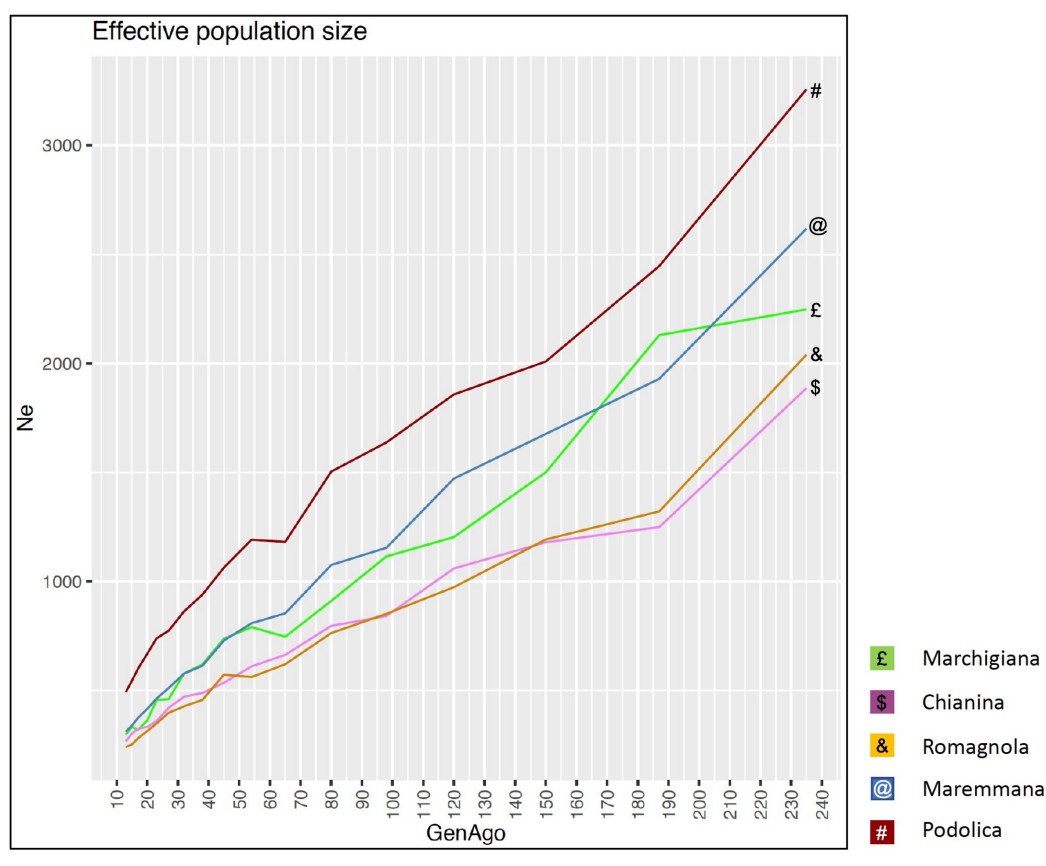

**Figure 6** Effective population size ($N_e$) across generation (GenAgo) for each Italian beef cattle breed.

10 years and, to the best of our knowledge no genetic bottlenecks have been recorded. Interestingly, *Weigel (2001)* also observed a lack of correlation between the magnitude of inbreeding and the population size of five major US dairy breeds (Ayrshire, Brown Swiss, Guernsey, Holstein, and Jersey). Noteworthy, breeds with small (Jersey) and large (Holstein) population sizes had comparable levels of inbreeding, probably because in both populations the intensity of selection was considerably high (*Weigel, 2001*). We have compared the $F_{hat2}$ values measured in MAR, ROM, CHI, MRM, and POD breeds with $F_{hom}$ values reported by *Mastrangelo et al. (2018)* in the same breeds. As previously explained, these two inbreeding coefficients, generated with the PLINK v1.9 software (*Purcell et al., 2007*) are quite comparable and both measure the reduction in heterozygosity associated with inbreeding. We observed that in general the $F_{hom}$ coefficients measured by *Mastrangelo et al. (2018)* in the same five breeds analyzed by us are higher (in the 0.066–0.118 range) than the $F_{hat2}$ coefficients estimated in the current work. *Biscarini et al. (2020)* reported an $F_{ped}$ coefficient for the MRM breed of 0.049, which is similar to the result obtained by us (Table 2). In another study, *Moioli, Napolitano & Catillo (2004)* described Wright $F_{is}$ coefficients of 0.106 and 0.138 in the POD and MRM breeds, and *Santana Jr et al. (2012)* measured an $F_{ped} = 0.013$ in Brazilian MAR cattle. One potential reason for the discrepancy
between our results and those of *Mastrangelo et al. (2018)* is that we have estimated allele and genotype frequencies with much more accuracy because our sample size is much larger *i.e., Mastrangelo et al. (2018)* used samples sizes of 21–24 animals while we have used sample sizes of 334-909 animals. This can be particularly important when measuring the excess of homozygosity in SNP markers with low or very low minimum allele frequencies.

Despite the fact that we have detected fluctuations in the magnitudes of $F_{hat2}$ and $F_{ped}$ during the last 20 years, these oscillations were not very important (Figs. 2 and 3). Changes in homozygosity and inbreeding coefficients across time might be partly explained by the fact that the number of breeders, and particularly sires, is not constant across years. We have also observed a slight increase of $F_{hat2}$ in the five breeds during 2002–2020, particularly in MRM, POD, and CHI, increasing 0.01–0.02% per year, while $F_{ped}$ remained quite stable, with an annual rate of increase of 0.003–0.004% per year (Figs. 4 and 5). *Mc Parland et al. (2007)* measured the evolution of the inbreeding in five Irish cattle and observed that in the 1994–2004 period $F_{ped}$ remained constant or decreased in Angus, Charolais, and Limousine populations, while in Hereford, Holstein-Friesian, and Simmental a yearly increase of $F_{ped}$ (between 0.06–0.13%) was detected. In another study, *Weigel (2001)* also reported increases of $F_{ped}$ of 0.10–0.15% per year. In the case of the five Italian breeds under study, such increases are much more modest, because the ANABIC genetic program is designed to minimize inbreeding. A high number of bulls is used in natural reproduction, rather than a few elite sires, and each one of them is mated with few cows, because this approach ensures an offspring with low inbreeding coefficients. The matings are programmed and designed to minimize inbreeding, using the less related pairs of breeders and allowing a maximum inbreeding increment of 5% in each mating (*Sbarra, 2011*).

By using the SNeP (*Barbato et al., 2015*) and Endog v4.8 (*Gutiérrez & Goyache, 2005*; *Gutiérrez, Goyache & Cervantes, 2009*) tools, we have detected a sustained and marked decline in the effective population size of the five breeds under investigation. *Mastrangelo et al. (2018)* reported a very similar tendency in the same five populations. Effective population size is a complex parameter defining the size of a Wright–Fisher ideal population generating the same rate of inbreeding and variance of gene frequencies detected in the real population under investigation (*Crow & Kimura, 1970*). In principle, selection and reproductive management, particularly in AI schemes in which a reduced number of bulls mate with a large number of cows, are expected to reduce $N_e$. Although, $N_e$ cannot be equated to a coefficient of inbreeding or diversity because it depends on many variables (*Wang, Santiago & Caballero, 2016*), it is remarkable that this strong $N_e$ decline (Fig. 6) was not accompanied by a substantial reduction in *Ho* or *He* in the last 20 years (Figs. 2 and 3). Increased variance in family size associated with the upward trend to use a reduced number of elite sires as breeders could be one of the major reasons for this progressive decline of $N_e$. However, it should be also noticed that estimates of $N_e$ historical trajectories with the SNeP tool (*Barbato et al., 2015*) are sometimes unreliable, particularly in the most recent and oldest generations (*Corbin et al., 2012*). Besides, $N_e$ estimates are strongly affected by data manipulation factors (*e.g.,* choice of the minimum allele frequency threshold) employed in the analysis (*Corbin et al., 2012*; *Barbato et al., 2015*) and one of the main tenets of the

coalescent is that no SNP can be reliably sampled after $4N_e$ generations in the past. So, our estimates of $N_e$ historical trajectories should be interpreted with caution.

## CONCLUSIONS

The low level of inbreeding found in this study confirms the success of the Italian beef cattle selection program carried out by ANABIC, which aimed to minimize inbreeding. We have observed that the annual rate of increase of inbreeding in the five Italian cattle under study are lower than what has been reported in several dairy breeds from the United States of America and Ireland, probably because of factors related with reproductive management (high number of breeding bulls, matings programmed to minimize inbreeding, etc.). We have also detected a strong decrease of the effective size that is not accompanied by marked reductions of diversity or substantially increased inbreeding. These $N_e$ estimates should be interpreted with caution due to the inherent difficulty of measuring this complex parameter in a reliable manner.

## ACKNOWLEDGEMENTS

The authors wish to thank the ANABIC (S. Martino in Colle, Perugia, Italy) for their cooperation in this study, selecting and providing SNPs data and phenotypic information of Italian beef cattle breeds. The authors also wish to thank the editor and the three referees for their valuable comments on the manuscript and their constructive suggestions.

### Funding

This work was supported by the projects: ''Italian Biodiversity Environment Efficiency Fitness''—I-BEEF 1 and 2 —2014–2020 and 2020–2023. PSRN: Support for the conservation, use and sustainable development of genetic resources in agriculture (Sub-measure 10.2). The funders had no role in study design, data collection and analysis, decision to publish, or preparation of the manuscript.

### Grant Disclosures

The following grant information was disclosed by the authors:
Italian Biodiversity Environment Efficiency Fitness''—I-BEEF 1 and 2 —2014-2020 and 2020-2023.
PSRN.

### Competing Interests

The authors declare there are no competing interests.

### Author Contributions

- Giacomo Rovelli performed the experiments, analyzed the data, prepared figures and/or tables, authored or reviewed drafts of the paper, and approved the final draft.

- Maria Gracia Luigi-Sierra and Dailu Guan performed the experiments, analyzed the data, authored or reviewed drafts of the paper, and approved the final draft.
- Fiorella Sbarra performed the experiments, authored or reviewed drafts of the paper, and approved the final draft.
- Andrea Quaglia and Francesca Maria Sarti analyzed the data, authored or reviewed drafts of the paper, and approved the final draft.
- Marcel Amills and Emiliano Lasagna conceived and designed the experiments, authored or reviewed drafts of the paper, and approved the final draft.

## Animal Ethics

The following information was supplied relating to ethical approvals (i.e., approving body and any reference numbers):

Blood extraction from young bulls is a routine procedure performed by trained veterinarians from the National Association of Italian Beef Cattle Breeds (ANABIC) for purposes that are completely unrelated to this research project. These blood samples were retrieved well before (2002–2019) our research project began and they were kept by ANABIC.

The data were collected in accordance with the FAO guidelines for the characterization of animal genetic resources. Animal management, animal husbandry and trait recording followed the criteria for the assessment of animal welfare as identified and defined by the Welfare Quality Project (*Vapnek & Chapman, 2011*; *Food & Agriculture Organization of the United Nations FAO, 2012*).

In summary, ethical permission from the University of Perugia was not required.

## Data Availability

All the data supporting the results are available in the article or in the Supplemental Files.

The raw genotypic and pedigree data available at figshare: Rovelli, Giacomo; Luigi Sierra, Maria Gracia; GUAN, DAILU; Sbarra, Fiorella; Quaglia, Andrea; Sarti, Francesca Maria; et al. (2021): Inbreeding trends in five Italian beef cattle breeds: a perspective of almost two decades (2002-2020). figshare. Dataset. https://doi.org/10.6084/m9.figshare.14625864.v1.

## Supplemental Information

Supplemental information for this article can be found online at http://dx.doi.org/10.7717/peerj.12049#supplemental-information.

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
