# Peer review of "Evolution of inbreeding: a gaze into five Italian beef cattle breeds history"

_PeerJ, doi:10.7717/peerj.12049_

## Round 0.1 · original submission · Major Revisions

1.The title of the study is not appropriate with regard to the study, should be modified.

2. Give the details of tools and pipelines used for  Chip-seq data analysis.
If more than one tool had been used for genotypic data analysis, the results would have been more reliable.  Like the use of GWAMA etc.

3. Details data filtrating, data augmentation are missing.

4. Why inbreeding coefficient wasn't calculated using BLUP?

5. Can I know if there was missing information in phenotypic data if yes, how did you manage that?

6. Authors write  "the annual rate of increase of inbreeding in the five Italian cattle under study are lower than what has been reported in several dairy breeds from the United States of America and  Ireland". I want the statement to be elaborated in discussion in light of accurate figures.

Reviewer 1 ·

Basic reporting

The research presented is interesting, well written and correctly performed.

Experimental design

N/A

Validity of the findings

Conclusions directly derive from results. In turn, results are consistent with expectations and data available.

Additional comments

Manuscript PeerJ #61227 titled “Inbreeding trends in five Italian beef cattle breeds: a perspective of almost two decades (2002-2020)” by Rovelli et al.
I enjoyed reading this paper. The research presented is interesting, well written and correctly performed. I have no doubts in recommending the publication of this paper in PeerJ as-is.
However, with the aim of improving the interpretation of the research presented, I would like to raise a few comments that can be considered or not by the authors.
L80-81: you stated that “Our goal was to test whether intensive selection performed in the last twenty years has resulted in a significant increase of inbreeding levels in these five populations”. However, you do not present examples illustrating selection programmes involving the cattle breeds analysed (e.g. Sarti et al., J Anim Sci., 2013, 91; Sbarra et al., Ital. J. Anim. Sci. 2009,8; Fioretti et al., 2020, Livest. Sci., 232; just examples). Probably, the intensity of selection in the five breeds is not comparable and it may be lower in Maremmana and Podolica cattle (not sure). Perhaps a wider explanation of the goal (selection intensity and, e.g., temporal trends) could be advisable.
L200-209: yes, I absolutely agree. This is a clear case of population structuring due to multiple (within herd) founder effects causing that, overall, inbreeding exceeds coancestry. If between-lines (herds) gene flow is promoted this would cause a quick drop of overall inbreeding and, at the molecular level, a heterozygote excess in subsequent few generations. However, the discrepancies between Fhat and Fped may deserve an additional (indeed short) comment. Fhat is closely related to Fis and (also in the case of pedigree-computed Fis) can be interpreted in breeding policy terms: positive and high values mean that mating between close relative are not –or cannot be- avoided). Perhaps, in the smaller populations breeders are performing a more strict breeding policy to limit within-herd inbreeding.

·

Basic reporting

I enjoyed reading the present paper and suggestions enclosed below are only suggestion as they may confer the paper with additional interesting information. However, I truly think the paper is well written and already provides a good quality material that is definitely worth being published.

Line 52 move from the same population before extracted at random.

Line 53 remove in order.

Line 59 to 61. Try not to make this expression this sharp as the topic raises controversy. The truth is that some recent papers have revealed the distortion caused by ancient contributions to inbreeding dilutes up to a degree in which pedigree-based analyses and genomic analyses may not differ that much. This should be acknowledged.

Line 68 do you mean effective population size?

Line 92. How many generations were presumably tested? Please clarify. Completeness index level (5th first generations) of the pedigree used should be provided if it is available. Endog calculates it.

M&M is well defined and clear enough as to ensure the replicability of this study. I particularly enjoyed reading the statistical approach used for the data.

Endog also permit to calculate Ne via different methods, such as individual increase in inbreeding, coancestry as the most accurate. Have the authors tried to calculate it and compare the values with those derive from genomic analyses?

Experimental design

Appropriate, see comments above

Validity of the findings

Appropriate, see comments above

Additional comments

I enjoyed reading the present paper and suggestions enclosed below are only suggestion as they may confer the paper with additional interesting information. However, I truly think the paper is well written and already provides a good quality material that is definitely worth being published.

Reviewer 3 ·

Basic reporting

The authors analyzed the inbreeding and diversity parameters in Italian dairy cattle, for four decades.

Experimental design

scientifically sound

Validity of the findings

Findings have great validity.

Additional comments

none

---

## Round 0.2 · accepted · Accept

I recommend the manuscript for publication in PeerJ.

Reviewer 1 ·

Basic reporting

All my concerns have been correctly addressed and, therefore, I'm happy to recommend the acceptation of this paper for publication in PeerJ.

Experimental design

N/A

Validity of the findings

N/A

Additional comments

I'm happy with the revision presented.